# Challenges and Opportunities in Managing Geriatric Depression: The Role of Personalized Medicine and Age-Appropriate Therapeutic Approaches

**DOI:** 10.3390/pharmaceutics16111397

**Published:** 2024-10-30

**Authors:** Agnieszka Jaros, Filip Rybakowski, Judyta Cielecka-Piontek, Magdalena Paczkowska-Walendowska, Bogusław Czerny, Adam Kamińki, Rasha Wafaie Mahmoud Elsorady, Agnieszka Bienert

**Affiliations:** 1Department of Clinical Pharmacy and Biopharmacy, Poznan University of Medical Sciences, 60-806 Poznan, Poland; agnieszkajaros0@gmail.com; 2Head of Adult Psychiatry Clinic, Poznan University of Medical Sciences, 60-810 Poznan, Poland; filrybak@ump.edu.pl; 3Department of Pharmacognosy and Biomaterials, Faculty of Pharmacy, Poznan University of Medical Sciences, 3 Rokietnicka St., 60-806 Poznan, Poland; jpiontek@ump.edu.pl (J.C.-P.); mpaczkowska@ump.edu.pl (M.P.-W.); 4Institute of Natural Fibers and Medicinal Plants National Research Institute, ul. Wojska Polskiego 71 b, 60-630 Poznan, Poland; boguslaw.czerny@pum.edu.pl; 5Departament of General Pharmacology and Pharmacoeconomics, Promeranian Medical University in Szczecin, 71-210 Szczecin, Poland; 6Department of Orthopedics nad Traumatology, Independent Public Clinical Hospital No. 1, Promeranian Medical University in Szczecin, Unii Lubleskiej 1, 71-252 Szczecin, Poland; adam.kaminski@pum.edu.pl; 7Head of Clinical Pharmacy Departments at Alexandria University Hospitals, Alexandria University, Alexandria 21523, Egypt; rasha.elsorady@gmail.com

**Keywords:** geriatric depression, personalized medicine, pharmaceutical forms

## Abstract

The global aging population has experienced rapid growth in recent decades, leading to an increased prevalence of psychiatric disorders, particularly depression, among older adults. Depression in the geriatric population is often compounded by chronic physical conditions and various psychosocial factors, significantly impacting their quality of life. The main question raised in this review is as follows: how can personalized medicine and age-appropriate therapeutic approaches improve the management of geriatric depression? This paper explores the epidemiology of geriatric depression, highlighting the influence of gender, race, and socioeconomic status on its prevalence. The classification and diagnosis of geriatric depressive disorders, based on ICD-11 and DSM-5 criteria, reveal the complexity of managing these conditions in older adults. Personalized medicine (PM) emerges as a promising approach, focusing on tailoring treatments to the individual’s genetic, clinical, and environmental characteristics. However, the application of PM in this demographic faces challenges, particularly in the context of pharmaceutical forms. The need for age-appropriate drug delivery systems is critical, given the prevalence of polypharmacy and issues such as dysphagia among the older patients. This study emphasizes the importance of developing patient-centric formulations to enhance the effectiveness of personalized therapy in geriatric patients.

## 1. Epidemiology

Over recent decades, there has been a notable escalation in the global older population, largely driven by a simultaneous decline in mortality and fertility rates, alongside improvements in quality of life. This trend has contributed to the extension of the average lifespan. According to World Health Organization (WHO) data, the number of people over the age of 60 is increasing. In 2019, the number of people over the age of 60 was 1 billion. It is estimated that this number will increase to 1.4 billion in 2030 and to 2.1 billion in 2050. The process of aging is associated with a decline in physical health and cognitive function, as well as an increased prevalence of psychiatric disorders. Depression exerts a detrimental influence on daily activities, and it closely correlates with the development of chronic conditions such as asthma, angina pectoris, arthritis, and diabetes. Also, the presence of numerous psychological factors, such as loss of partner, offspring, socioeconomic status, social role, physical and cognitive functioning, and occupational position diminishes the sense of security, thus correlating with the higher incidence of depression and psychotic disorders [1]. It has been demonstrated that individuals affected by depressive disorders have a 40% higher risk of premature death. In accordance with a WHO report from 2023, it is estimated that approximately 5.7% of people aged 60 and older experience depression [2,3]. Depression occurs 50% more frequently in women than in men (Figure 1). In older patients, the difference between genders is slightly reduced, especially among the oldest. Additionally, race and ethnic background may also influence the frequency of depression occurrence. Research suggests that depression may be more prevalent among older Latino women than in non-Latino women [4]. Comprehending the worldwide prevalence of depression among older adults is crucial for facilitating early detection and implementing effective therapy in this population [5,6]. Attention should be focused on the regions where populations, especially women and older women are exposed to stressful conditions. A new study by Alsous M et al. [7] has shown acute psychiatric repercussions of the Gaza war on Jordanian females. It turned out that 32.3% of them exhibited severe post-traumatic stress disorder (PTSD) symptoms, 53.4% had severe depressive symptoms, 48.2% reported severe insomnia, and 17.2% were diagnosed with fibromyalgia. In such circumstances, there is an alarming need of on-time psychiatric intervention. Diagnosing depression in older adults is considerably more challenging due to the difficulty in distinguishing depressive symptoms from nonspecific symptoms associated with aging and the presence of multiple medical comorbidities. Despite these challenges, depression in older patients is associated with increased morbidity and mortality [8].

## 2. Classification of Geriatric Depressive Disorders

Depression is diagnosed based on criteria outlined in the International Classification of Diseases, 11th Revision (ICD-11), and the Diagnostic and Statistical Manual of Mental Disorders, 5th Edition (DSM-5). In the geriatric population, we observed various types of depression, including major depressive disorder (MDD), minor depressive disorder (MnDD), dysthymic disorder, bipolar I disorder (most recent episode depressed), and adjustment disorder with depressive mood (Table 1).

Although the chronic depressive state in minor depressive disorders demonstrates a reduced intensity compared to that typically observed in major depressive disorders, the level of disability in both conditions is comparable [13]. The adjustment disorder describes an emotional reaction to an identified psychosocial stressor and encompasses individuals who manifest difficulty in adapting following a stressful event, disproportionate to the intensity or severity of the stressor [14].

## 3. Diagnosis of Depressive Disorders Including Distinctive Features in the Geriatric Population

The diagnostic criteria for all depressive disorders remain consistent across age groups yet diagnosing it in older adults poses increased complexity due to prevalent factors including general medical comorbidities, cognitive impairments, and adverse life events. Regardless of gender and specific medical conditions, the risk of depression increases with each acquired chronic illness [15,16]. Considering this, several scales for screening depression have been developed, focusing on older patients. Among these are the Geriatric Depression Scale (GDS), the Center for Epidemiologic Studies Depression Scale (CES-D), and the Beck Depression Inventory for Primary Care (BDI-PC). Within the geriatric cohort, the GDS is the prevailing scale in use. The full version of the scale comprising 30 questions (GDS 30) requires a significant amount of time for assessment. However, a shortened form containing 15 questions has also been approved and widely used [17,18]. It is an effective tool for simultaneously distinguishing among MDD, MnDD, and normal status within the geriatric population [19]. Distinctive clinical features observed in the geriatric population affected from depression are as follows:Patients often manifest comorbidities across various somatic conditions.Environmental and psychological factors exert heightened influence on older patients.The course of the illness is often atypical.Patients frequently experience adverse effects during pharmacological treatment [20].

Patients suffering from depression, lacking distressing symptoms like suicidal ideation or self-injury, may undergo ambulatory treatment. However, it is necessary to assess their ability to adhere to therapeutic recommendations and maintain treatment continuity, or the presence of a caregiver who can support the therapy process. Antidepressant medications represent an effective therapeutic option for mild, moderate, and severe episodes of depression. 

## 4. Therapy of Depression Including Aging-Related Aspects

Currently, the most commonly used group of medications includes new antidepressants, such as selective serotonin reuptake inhibitors (SSRIs)—fluoxetine, paroxetine, fluvoxamine, sertraline, citalopram, escitalopram—and selective norepinephrine reuptake inhibitors (SNRIs)—venlafaxine, duloxetine, levomilnacipran, desvenlafaxine [15,21].

### 4.1. SSRI

SSRI medications are favored as first-line treatment. They have predominantly replaced tricyclic antidepressants (TCAs) due to their superior adverse effects profile, reduced risk of overdose, and minimal anticholinergic activity [22]. However, they are not exempt from side effects. Typical gastrointestinal symptoms associated with SSRI use, including nausea, diarrhea, abdominal pain, and constipation, are predominantly linked to sertraline intake. Moreover, dry mouth may arise with paroxetine and sertraline, whereas headaches and decreased appetite are more commonly linked to fluoxetine intake. A meta-analysis of 51 randomized clinical trials did not reveal significant differences in efficiency among individual SSRIs [21]. The decision regarding the choice of a particular medication should be based on a thorough consideration of the patient’s specific characteristics as well as those of the medications (Table 2).

In randomized controlled trials, sertraline, paroxetine, and fluoxetine have been shown to be more effective than placebo in reducing symptoms of depression and increasing rates of remission. The response to SSRIs (defined as a reduction in the severity of depression symptoms by ≥50%), varied between 35% and 60%, compared to the placebo response rate rating from 26% to 40% [21,22,23,24]. It is important to recognize that while selective serotonin reuptake inhibitors (SSRIs) are generally regarded as safe for use in the geriatric population, caution is advised in individuals with renal impairment due to the risk of hyponatremia [20]. This phenomenon has been documented across nearly all SSRIs, as well as the serotonin and noradrenaline reuptake inhibitor (SNRI) venlafaxine. Extra care should be taken when co-prescribing medications alongside diuretics [23]. Additionally, the use of serotonin reuptake inhibitors (SRIs) should be approached carefully in patients at heightened risk of adverse cardiovascular events, as there is an association between this class of medications and an increased risk of sudden cardiac death (SCD) [24].

### 4.2. SNRI

Only 40 percent of patients experience a comprehensive response to the initial therapy agent employed [25]. If first-line treatment proves ineffective, the consideration of second-line medication, such as SNRIs, is advised. They dually inhibit serotonin and norepinephrine reuptake pumps, enabling the treatment of a broad spectrum of depressive symptoms. Among the most common adverse effects are anxiety, insomnia, headaches, and sexual dysfunction. In contrast to the SSRI class, the SNRI more frequently induces nausea, insomnia, dry mouth, and in rare instances increases blood pressure [26]. In a randomized study conducted by Allard et al., venlafaxine extended release was compared to citalopram in a parallel group of patients aged 65 and older. No statistical differences in efficacy were observed between the two groups [27]. Raskin et al. confirmed the superiority of duloxetine over placebo in their study and demonstrated good tolerability in patients aged 65 and above [28].

### 4.3. TCA

Currently, TCAs are commonly prescribed for off-labels uses [29]. Despite the increasing popularity of selective serotonin reuptake inhibitors and other newer antidepressants, physicians might still prescribe TCAs when conventional therapy has proven ineffective. Although, Mi-ju Park et al. demonstrated in 2016 that the number of prescribed TCAs decreased from 80,217 to 72,287 compared to 2014. According to the Beers Criteria, TCAs are potentially inappropriate for older adults due to their strong anticholinergic effects, cardiotoxicity, and cognitive impairment [29,30]. This group includes amitriptyline, clomipramine, imipramine, nortriptyline, doxepin, opipramol, desipramine [31]. Dezepamine and nortriptyline are commonly used in this class of medications. They exhibit fewer anticholinergic side effects compared to amitriptyline, doxepin, and imipramine, which are generally recommended to be avoided in older patients [25].

### 4.4. Comparison of Drugs from Different Pharmacological Groups According to Systematic Reviews and Meta-Analyses

A recently published systematic review and network meta-analysis compared 21 antidepressants from different pharmacological groups, comparing their efficacy and acceptability in adult patients with major depression [32]. Although all examined drugs were more effective than placebo, some differences were found between specific drugs. However, one of the limitations of the analysis regarding geriatrics is the lack of data dedicated to older patients, as a specific population. On the other hand, a systematic review, pairwise and network meta-analysis dedicated to older patients published in 2019 [33], points to duloxetine and quetiapine as the drugs which have showed higher superiority to placebo than other antidepressants. However, the authors mention the limitation of the analysis, which results in a need to conduct further studies with better reporting of important characteristics, especially comorbidities and polypharmacy [33].

## 5. Pitfalls in Managing Pharmacotherapy

### 5.1. Age-Related Differences in PK and Consequences of Polypharmacy

In the geriatric population, depression is often incorrectly diagnosed, leading to suboptimal or inappropriate treatment. The effectiveness and safety of pharmacotherapy differ in older adults compared to younger adults. These disparities encompass physiological age-related changes affecting pharmacodynamics and pharmacokinetics, a high prevalence of comorbidities, an elevated risk of drug interactions due to polypharmacy, and variations in life circumstances [34]. These pharmacokinetic differences include variations in drug absorption, distribution, metabolism, and excretion mechanisms, which include:Absorption—In older patients, drug absorption does not significantly change with age.Distribution—As a consequence of alterations in adipose tissue composition, the distribution of lipophilic medications may be augmented in elderly subjects.Elimination—Hepatic metabolism and renal function gradually decline with age, limiting the elimination of certain drugs, especially those cleared through hepatic metabolism.

Numerous interactions between antidepressants and other medications predominantly occur during metabolic processes, primarily mediated by the hepatic CYP system and, to a lesser degree, the uridine diphosphate glucuronosyltransferase system. The elevated risk of adverse drug events (ADEs) may stem partly from dosing regimens that overlook age-related alterations in the pharmacokinetics of antidepressants and/or interactions between drugs [35]. Due to the increased likelihood of polypharmacy in geriatric patients when selecting therapy, it is important to consider the potential for drug–drug interactions. In a 2011 study involving 877 nursing home patients, it was found that up to 43.1% of prescribed antidepressant medications were potentially inappropriate. Particularly problematic was the dosing adjustment (observed in 8.8% of patients), while drug–drug interactions affected as many as 25.9% [36]. Physicians and clinicians often encounter difficulties in accessing the scientific literature regarding pharmacokinetics and drug interactions due to limited data provided in drug labels. The analysis conducted by Richard D. and colleagues revealed a significant disparity between pharmacokinetic interactions reported in the scientific literature and those documented in drug labels. It was found that the scientific literature documents nearly four times as many interactions affecting parameters such as AUC (area under the concentration–time curve) or Cl (clearance)—47 instances—compared to only 12 interactions included in accompanying labels by drug manufacturers. Furthermore, their research identified twelve antidepressants showing evidence of both age-related decreases in clearance and at least one pharmacokinetic interaction that further reduces clearance. It is essential for clinicians to recognize that when multiple factors contribute to reducing drug clearance in elderly patients, the likelihood of experiencing adverse drug events increases [30,31,32,33]. Therapeutic drug monitoring (TDM) can be a significant tool in treating depression in this patient group, allowing for dose adjustment to individual needs and minimizing adverse effects. It involves analyzing drug concentrations in serums or other body fluids that correlate with optimal therapeutic efficacy and minimal risk of adverse effects, enabling the adjustment of drug dosage to individual patient needs. TDM also facilitates the understanding of drug interactions, reasons for treatment non-response, and the impact of genetic variability. In response to these challenges, there is an increasing emphasis on individualizing therapy and utilizing methods such as therapeutic drug monitoring (TDM), which can help optimize treatment and minimize adverse effects. Despite the potential benefits of utilizing TDM in the geriatric population [37,38,39], Hermann et al. observed that the frequency of TDM usage decreases with age. They noted that TDM occurred three times less frequently in patients over 90 years old compared to patients in the youngest age group [40]. In a study conducted in Norway analyzing 35,000 serum concentration measurements obtained from patients utilizing TDM services, it was observed that the concentration-to-dose ratio (C/D ratio) for the studied antidepressants began to increase between the ages of 44 and 55. For citalopram, escitalopram, venlafaxine, and mirtazapine, the C/D ratio increased by 100% in the age range of 79–90 years. However, for sertraline, no significant changes in the C/D ratio were observed, suggesting that the patient’s age has little impact on the pharmacokinetics of sertraline. Presently, out of the antidepressants examined, only citalopram and escitalopram have dosage reduction guidelines for older adults incorporated into the Summary of Product Characteristics (SPC). In 2011, a suggestion was issued to halve the daily doses of citalopram/escitalopram for individuals aged 65 and older, prompted by findings from post-market surveillance regarding cardiac toxicity and growing apprehension regarding the dose-dependent risk of QT interval prolongation [41]. According to Tveit and colleagues, in comparison to 2007, a slight reduction in prescribed doses of antidepressants among older individuals in Norway was observed in 2017. A reduction in the proportion of older individuals with serum concentrations above the recommended reference range was only found for mirtazapine and individuals aged 80 years and above using venlafaxine. For the most used antidepressants, citalopram and escitalopram, prescribed doses were slightly reduced (by 10–15%), but the proportion of patients with serum concentrations above the recommended reference range remained unchanged. The overall findings of this study suggest that a significant proportion of older individuals still attain high serum concentrations of antidepressant medications [38].

### 5.2. Adherence to the Therapy

Therapeutic drug monitoring (TDM) can also be used when patients fail to adhere to therapeutic recommendations. Noncompliance with these instructions results in significant fluctuations in plasma drug concentrations, which have been shown to correlate with adverse clinical outcomes [25]. Strategies aimed at improving adherence to recommendations among geriatric patients should consider various factors, including comorbidities, age-related cognitive factors, as well as environmental and social factors. Additionally, these strategies should be tailored according to various depression parameters, such as symptom profile, subtype, disease severity, age of onset, presence of suicidal ideation, and prior experience with antidepressant medications. There are numerous strategies aimed at enhancing patient adherence in depression treatment. These include modifying existing therapeutic practices and promoting broader education, not only for patients themselves but also for patient caregivers, healthcare providers, and public health entities. By increasing the understanding and awareness of depression and offering suitable resources and assistance, it is feasible to reduce the stigma linked with the condition and improve the effectiveness of treatment [42].

### 5.3. Delayed Response to Antidepressant Treatment

The delayed onset of antidepressant effectiveness presents a hurdle in promptly determining the right dosage. In older adults, treatment response might lag that of younger individuals. Research from 2008 highlighted that longer trial durations (10–12 weeks) substantially increased the likelihood of treatment response among older adults [43]. While some improvement may appear within 4–8 weeks, it usually requires 2–3 months of therapy to unlock the full therapeutic benefits [21].

One of the contributing factors is the interpretation of depressive symptoms as an outcome of the aging process (limitations imposed by functional disability, decreasing social contacts, going on retirement, and grief) or as a common response to the presence of comorbidities [44]. Another issue lies in the specificity of depressive symptoms in elderly patients. They frequently present less specific symptoms, including insomnia, weight loss, fatigue, and headaches, which can overlap with or be confused with other physical illnesses and dementia. Concurrently, older individuals may attribute their symptoms to a somatic illness, thereby evading appropriate medical assistance [25,45].

### 5.4. Therapy Resistance

Treatment-resistant depression (TRD), a subset of major depressive disorder, is defined by its unresponsiveness to conventional first-line therapeutic approaches. Despite this, no improvement is observed following the administration of two distinct classes of antidepressant medications for a minimum duration of 8 weeks. TRD is prevalent in the geriatric population suffering from depressive disorders [46,47]. The evidence base for treating TRD is limited and includes the following:Switching medications within the same class (e.g. from one SSRI to another)Switching medication to a different class (e.g. SSRI to TCAs)Increasing the dosageAdding a second antidepressant from a different class [21].

After any treatment change, close monitoring of the patient is essential. If there is no moderate improvement after an additional 4–8 weeks of treatment, a comprehensive psychiatric reassessment should be conducted [21]. In cases of treatment failure with SSRI and SNRI, bupropion therapy may be considered, particularly in patients experiencing fatigue, sexual dysfunction, and weight gain [48]. However, for patients experiencing anxiety states and insomnia, a reasonable option is augmentation with mirtazapine [49]. Recently, to increase the effectiveness of therapy of depression as well as to achieve faster onset of treatment, new adjuvants out of the mentioned antidepressant groups have been proposed and studied, i.e., NMDA antagonists (esketamine), nitrous oxide modulators, anti-inflammatory drugs, and probiotics [50].

## 6. Personalized Medicine

Current psychiatric medications are quite effective, but many patients still fail to achieve remission. Knowledge of which patient will respond to which pharmacological medication remains unknown and often relies on the current selection of medications through trial and error [51,52]. Personalized medicine (PM) is founded upon the recognition of variances among patients suffering from the same illness and utilizes the unique clinical, genetic, and environmental features of each patient. According to this approach, the goal is to tailor appropriate therapies to specific patient groups. PM enables the prediction of the effectiveness of a given therapy for an individual patient [52,53]. There are numerous benefits to the use of personalized medicine:The reduction of side effects;Identifying the best treatment options based on patient characteristics;Enabling better disease prevention;Lowering healthcare costs;Increasing patient engagement;Promoting research and innovation [53].

Personalized medicine holds the promise of identifying genetic alterations that either increase susceptibility to diseases or offer protective factors against them [54]. MDD has a strong genetic component, with an estimated risk of developing MDD ranging from 40 to 70% [55]. In recent years, there has been a significant effort to identify and propose various biological markers of depression, such as genetic, metabolic, or inflammatory factors. The aim has been to understand variations in the course of the illness, identify prognostic factors, and better tailor treatment approaches [56]. The clinical significance of pharmacogenetic factors in the pharmacokinetics and pharmacodynamics of antidepressants is currently increasingly recognized [34]. Current data suggest that pharmacogenomic-guided antidepressant prescribing improves the treatment of depression. However, the benefit has not been fully established yet and needs to be further studied, taking into account different patient populations [57,58]. Personalized medicine in the geriatric population needs to take into account the pitfalls of the managing therapy of depression in older patients, discussed in chapter 5 of this manuscript.

### 6.1. Genetic Factors Impacting Drugs PK and PD

According to the Clinical Pharmacogenetics Implementation Consortium (CPIC) in 2023, polymorphisms in genes encoding the enzymes CYP2C19, CYP2D6, and CYP2B6 may significantly influence the pharmacokinetics of antidepressant medications and the recommended dosage. However, when it comes to genetic variability in pharmacodynamics, the data are not as clear-cut. In scientific reports suggesting the role of genes such as SLC6A4 (encoding the serotonin transporter) and HTR2A (encoding the serotonin-2A receptor), it cannot be ruled out that genetic variability in pharmacokinetics may be one of the possible reasons for the observed relationships at the pharmacodynamic level [59].

The advent of genomic techniques has provided the foundation for personalized medicine in neuropsychiatric conditions, such as major depression. It has become evident in recent years that variations in several genes governing pharmacokinetics and pharmacodynamics may play a role in both the progression of depressive episodes and the response to antidepressant therapy [60].

In pharmacogenetic studies of antidepressant drugs, significant attention has been devoted to the genetic variability of the serotonin transporter gene (5-HTT), located on chromosome 17q12 in humans [61]. The serotonin transporter (5-HTT) is a protein tasked with the reabsorption of serotonin from the synaptic space into presynaptic neurons. It serves as the primary focus of selective serotonin reuptake inhibitors (SSRIs), the current preferred class of antidepressant medications. Within the promoter region of the serotonin transporter gene (SLC6A4), there exists a serotonin transporter-linked polymorphic region (5-HTTLPR), featuring long (l) or short (s) alleles. These alleles correspond to varying levels of SLC6A4 gene activity, with the long allele associated with higher activity and the short allele linked to lower activity [62]. It is postulated that carriers of the l allele may exhibit a better response to antidepressant treatment due to increased expression/activity of 5-HTT, which is a key target for the action of most of these drugs [63]. It was demonstrated that older patients with depression who were homozygous for the l allele responded faster (as early as the second week) to paroxetine treatment compared to carriers of the s allele. Furthermore, the lack of influence of the 5-HTTLPR polymorphism on the onset of therapeutic response to nortriptyline suggests that geriatric patients with increased serotonin uptake may exhibit more acute reactions to SSRIs [64]. The meta-analysis conducted by Porcelli et al. revealed an association between the 5-HTTLPR polymorphism and treatment outcomes after stratifying the results by ethnicity. The authors confirmed that the l/l genotype was associated with a better response to SSRIs, particularly in the Caucasian population [65].

### 6.2. Biomarkers of the Susceptibility to Depression and the Response to Treatment

The FDA-NIH Biomarker Working Group defined biomarkers as “a defined characteristic that is measured as an indicator of normal biological processes, pathogenic processes or responses to an exposure or intervention” [62]. Early research on biological markers of antidepressant treatment response suggests that levels of urinary 3-methoxy-4-hydroxyphenylglycol may predict a differential response to adrenergic and serotonergic medications, but these findings have not been confirmed [66]. Strong evidence supports the role of hypothalamic-pituitary-adrenal (HPA) axis dysregulation in the pathophysiology of adult depression. Approximately half of adult patients with depression exhibit stress-induced HPA axis hyperactivity alongside impaired negative feedback. This dysregulation manifests as persistent hypercortisolemia and failure to suppress cortisol levels in the dexamethasone suppression test (DST). Studies found that patients who did not respond to DST were typically older, had more severe symptoms, more frequently exhibited psychotic features, and were significantly less responsive to placebo compared to DST responders. However, they did not show increased responsiveness to tricyclic antidepressants or electroconvulsive therapy, suggesting that this abnormality simply indicated a greater need for active treatment [67]. Moreover, studies consistently show that an elevated cortisol response to the DST is associated with more severe depressive symptoms, suggesting it as a marker of clinical severity [68].

The neurotrophic theory of depression posits that environmental stress factors and mutations decrease BDNF synthesis in the brain, leading to reduced synaptic plasticity, decreased synaptic transmission, and increased neuronal degeneration. These impairments are considered to be the cause of specific structural changes in brain regions known to be involved in cognition and mood regulation, such as the atrophy of the prefrontal cortex and shrinkage of the hippocampus. This theory is supported by studies demonstrating decreased levels of BDNF in post-mortem brain samples from patients with MDD [69,70,71]. Considering the role of BDNF and its related receptors in neural plasticity and structural changes, along with the fact that depression and antidepressants exert opposing effects on BDNF expression and TrkB function, it is evident that BDNF may play a pivotal role in the pathophysiology of MDD and the mechanism of action of antidepressant drugs. Research described in this review clearly indicates that both early-onset and late-onset depression are associated with altered BDNF expression. As discussed earlier, BDNF and TrkB expression decreases with age [72].

Several studies have explored the relationship between genetic polymorphisms in the norepinephrine transporter (NET) gene SLC6A2 and susceptibility to depression, yielding inconclusive results. Nevertheless, certain NET variants have been shown to affect the response to antidepressants. In a study conducted in Korea, the G/G genotype of the G1287A polymorphism in exon 9 of the NET gene was found to positively correlate with a favorable response to nortriptyline in elderly individuals, but it had no effect on the response to SSRIs such as fluoxetine and sertraline [73]. Pharmacogenetics has played a significant role in identifying functional polymorphisms in key drug-metabolizing genes [65]. Particularly, considerable attention has been devoted to the CYP and MDR1 (multidrug resistance 1) gene families in the context of drug metabolism studies. Research on the CYP2D6 enzyme (debrisoquine/sparteine hydroxylase) has gained substantial popularity compared to other CYP isoforms, as it is involved in the metabolism of many commonly prescribed antidepressant drugs (e.g., fluoxetine, paroxetine, and clomipramine) [74,75]. A pharmacokinetic analysis conducted by Feng et al. revealed that elderly depressed patients with different CYP2D6 genotypes exhibit varying drug elimination rates, which may necessitate dose adjustments based on metabolizer status [75]. Changes in the expression of genes encoding proteins involved in monoamine metabolism, such as 5-HTT or TPH, may serve as indicators of an increased risk of depression or resistance to antidepressant treatment [56].

Clinical risk factors can contribute to improved diagnostics, treatment selection, assessment of treatment response, and prediction of psychiatric illnesses. One example is the presence of severe depression in childhood or previous failures in psychiatric treatment, both of which are clinical risk factors for bipolar disorders [76,77]. Biomarkers are objective, measurable features of biological processes, often used to indicate pathology and/or pharmacological response to intervention. In psychiatry, there are three subcategories of biomarkers: molecular markers, genetic markers, and neuroimaging markers. Biomarkers have the potential to elucidate the etiology of clinical symptoms and attempts are made to classify psychiatric disorders based on biomarkers. Apart from the promising aspects of personalized medicine, there are apprehensions regarding the sharing of data, patient confidentiality, and equitable access to treatment [53].

Opportunities and challenges of personalized medicine are summarized in Table 3.

## 7. Pharmaceutical Forms for Personalized Therapy

Various dosage forms are currently available (e.g., tablets, injections, patches, nasal sprays), of which oral drug administration provided as solid oral dosage forms remains the major route of drug therapy, due to cheap manufacturing process, accurate dosing, patient acceptability, and taste masking [78]. There is convincing evidence that the population of older people is experiencing an increasing number of clinically significant swallowing problems (such as dysphagia), particularly when multimorbid, fragile, and polymedicated individuals are considered [79]. Swallowing disorders negatively impact the administration of conventional tablets, leading to poor compliance and inappropriate modifications (e.g. crushing, splitting) [80]. Nevertheless, to effectively improve oral formulation administration for older patient populations, novel formulation designs must be considered by using a patient-centric strategy. In addition to acceptable reductions in oral formulation size, novel film coating materials that can be applied to oral formulation and offer swallowing safety and effectiveness with minimal patient effort are still required [81]. However, despite the best pharmaceutical technology, conventional pills do not meet the needs of personalized therapy.

As part of such therapy, to bypass swallowing problems, orodispersible film (ODF) is worth considering. Orodispersible film (ODF) is a solid dosage form that might make oral drug distribution easier for these patients. ODFs are inserted into the mouth, and as they dissolve, the drug is taken together with saliva to enter the digestive system. ODFs can be readily chopped into pieces before administration or used during the production process to achieve flexible dosage. ODFs have been proposed as an appropriate dosage form for geriatric patients, particularly those with dysphagia [81]. Furthermore, ODF has many benefits, including the ability to be transported easily due to its small size and weight, a convenient dose form, precise dosing, the ability to be taken with or without water, and the elimination of the risk of choking [82]; while disadvantages are not noted. However, ODFs do not constitute a directly personalized therapy in the context of the drug content, but only the pharmaceutical form itself.

An interesting strategy for personalized therapy in the context of delivering multiple substances simultaneously is the polypill, which was developed over two decades ago [83]. It was developed to offer a therapeutic choice that would increase treatment results and patient adherence. The pill was a concoction of drugs that were already on the market to treat strokes, ischemic heart disease, and cardiovascular disease. With dosages intended to optimize each component’s efficacy, decrease side effects, and lower treatment costs, it was available as a single tablet to be taken once a day [84]. Their use is subject to certain restrictions. These include treating too-low blood pressure or LDL levels, worries about the use of needless drugs, and challenges in modifying the dosage to target patients. Another significant problem when modifying the dosage for aged patients, individuals with comorbidities, and patients currently taking numerous drugs is altering the titration of the different components in the polypill [85].

Since three-dimensional printing (3DP) started to be used in the manufacturing of different medication forms as well as different doses, it has been evident that this cutting-edge technology has the potential to significantly advance the field of personalized medicine (PM) [86,87]. The use of 3DP makes it possible to produce complicated structures or layers with distinct release profiles, different-shaped dosage forms, and formulations including several active ingredients [88,89]. Additionally, patient-specific dosages are simpler to obtain, especially for active ingredients with a limited therapeutic window, to account for variations in drug metabolism and tolerance. Furthermore, capable of disguising tastes and enhancing swallowability is 3DP [90]. Additionally, drug 3DP considers a wide range of factors, including filling percentage, shape/geometry, and heat treatment parameters of printed preparations, all of which may have an impact on the drug release rate [91].

Three-dimensional printing technology allows you to prepare one form containing several active substances [92]. This could be extremely useful for patients suffering from dementia who forget their medication as well as for caregivers [93]. It is easier to remember one pill in the morning compared to ten during the day at different times, which reduces the risk of errors [94]. It is worth emphasizing that poly 3DP drugs will be more suitable for patients who have already implemented a treatment regimen than for patients who are taking substances for the first time, since it is difficult to distinguish which substance may cause potential allergic reactions or side effects [95].

Importantly, 3DP can be adapted not only to conventional tablets. It enables the printing of various forms of modified-release drugs [96], as well as tablets or ODFs [97].

There is only one 3DP medication available at this time, Spritam, which was granted FDA approval in 2015 and is intended for people with epilepsy [98]. It makes use of ZipDose technology, a DOS variant. The medication dissolves more quickly in water thanks to its soluble matrix’s porosity, which makes it more effective than traditional tablets [99].

Although drugs can be 3D-printed, this technology has several significant advantages in personalized medicine, but it also has some disadvantages compared to current manufacturing methods. For example, the production efficiency in each time is much lower than in the case of conventional methods [95]. When opposed to a tablet machine, the speed at which a single tablet is created is slower. Additionally, this approach prints one tablet at a time; hence, additional printers would be needed for big runs. It is anticipated that the drop in costs, convenience of use, and technological advancements would lead to a greater usage of this production process [100]. Moreover, concerning 3D-printed medications, the regulations are currently unclear. The creation of committees and supporting initiatives could be seen as a positive step toward making 3DP more widely used in the pharmaceutical manufacturing industry [101]. Nevertheless, the absence of precise criteria for quality assurance, manufacturing processes, and legal procedures may deter businesses from making investments in the creation of 3D-printed products.

Overall, it is reasonable to assume that 3D technology will indirectly improve treatment effectiveness and outcomes in the geriatric population.

## 8. Conclusions

The aging global population necessitates a greater focus on the epidemiology and management of geriatric depressive disorders. Personalized medicine offers a significant advantage in tailoring treatments to individual patient needs, potentially improving therapeutic outcomes and reducing side effects. However, the success of PM in older populations is contingent upon the development of appropriate pharmaceutical forms that address the unique challenges of aging, such as polypharmacy and dysphagia. Future research should prioritize the design of patient-centric drug delivery systems to ensure the efficacy of personalized therapies in this vulnerable demographic. Also, new clinical trials should validate different approaches of personalized medicine in the geriatric population.

## Figures and Tables

**Figure 1 pharmaceutics-16-01397-f001:**
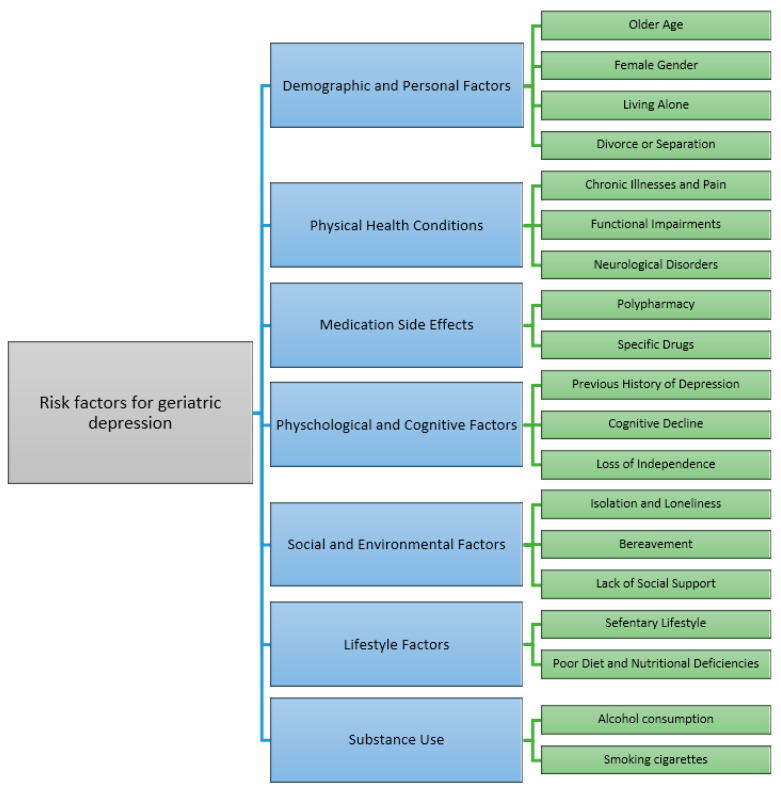
Risk factors for geriatric depression [9,10].

**Table 1 pharmaceutics-16-01397-t001:** Classification of geriatric depressive disorders according to DSM-5 [11,12].

Major Depressive Disorder	Minor Depressive Disorder	Dysthymia	Bipolar I Disorder, Most Recent Episode Depressed	Adjustment Disorder with Depressed Mood
Identification: at least five characteristic symptoms persisting for a minimum of two weeks, including: persistent depressed mood throughout the day, loss of interest or pleasure in activities, weight loss, sleep disturbances, chronic fatigue, difficulty concentrating, and presence of suicidal thoughts. These symptoms should not be directly caused by substance use, bereavement, or a medical condition [8].	Identification: at least two symptoms, including persistent depressed mood throughout the day, loss of interest or pleasure in activities, weight loss, sleep disturbances, chronic fatigue, difficulty concentrating, and presence of suicidal thoughts, but it cannot be diagnosed in individuals with a history of major depressive disorder, dysthymia, or bipolar disorder.	Dysthymia is characterized by a period (2 years or longer) of persistently depressed mood, present for most of the day, more days than not. During the 2-year period of the disturbance, the person has never been without symptoms from the above two criteria for more than 2 months at a time.	Currently (or most recently) in a major depressive episode. There has previously been at least one manic episode or mixed episode.	The development of emotional or behavioral symptoms in response to an identifiable stressor occurring within 3 months of the onset of the stressor. F43.21 With depressed mood: Low mood, tearfulness, or feelings of hopelessness are predominant.

**Table 2 pharmaceutics-16-01397-t002:** Factors influencing antidepressant medication selection decision [21].

Patient Features	Medication Features
Patients’ preference History of tolerance Comorbidities Concomitant medications	Side effects Cost Treatment regimen Safety in case of overdose Availability of different formulations

**Table 3 pharmaceutics-16-01397-t003:** Summary of opportunities and challenges of personalized medicine.

Challenges	Opportunities
Increase in risk factors resulting from demographic changes in society	New diagnostic methods, including scales and biomarkers, fast interventions in populations exposed to stressful conditions
The occurrence of treatment-resistant depression (TRD) and/or delayed onset of antidepressants	Switching medications within the same class (e.g. from one SSRI to another), Switching medication to a different class (e.g. SSRI to TCAs), Increasing the dosage, Adding a second antidepressant from different class Adding different adjuvants out of the antidepressant groups, like NMDA antagonist esketamine or other products including natural preparations
Nonadherence to treatment	Monitored therapy and multidisciplinary care
The need to adapt therapy to the individual patient	Reduction of side effects, Identifying the best treatment options based on patient characteristic, Enabling better disease prevention, Lowering healthcare costs, Increasing patient engagement, Promoting research and innovation New clinical studies on the effectiveness and utility of personalized medicine
Interindividual differences in PK and PD of antidepressants	TDM and PD biomarkers with pharmacogenetics
Difficulty in taking drugs resulting from, for example, dysphagia	New pharmaceutical forms dedicated to older patients, described in chapter 7

## Data Availability

All data supporting reported results can be found within the manuscript.

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
