# Peer review of "Challenges and Opportunities in Managing Geriatric Depression: The Role of Personalized Medicine and Age-Appropriate Therapeutic Approaches"

_pharmaceutics, 2024, doi:10.3390/pharmaceutics16111397_

Round 1
Reviewer 1 Report
Comments and Suggestions for Authors
The authors made great efforts to depict a comprehensive and exaustive state of the art of depression in elderly highlighting challenges and future possible direction to improve the management of that disease in a so special sb-population. I suggest to accept tha manuscript after minor revision will be made:
Line 117-118, remove "that" from the sentence "Meta analysis of 51 randomized clinical trials that did not reveal significant differences in efficiency among individual SSRIs."
Line 129 it could be better to report the references in the square brackets as [21-24]
Line 132, at the end of the sentence "....hyponatremia, the refence n. 20 was expected. If so, carefully review the numbered sequence of references in the main text
Line 201 [30,32,33], please modify as follows [30-33]
Line 210-211 "Therapeutic Drug Monitoring 210 (TDM)", remove Therapeutic Drug Monitoring leaving just TDM
Line 227 Correct the syllable division of the recommended word
Line 234 reference 78 probably is a mistake, verify.
Author Response
The authors made great efforts to depict a comprehensive and exaustive state of the art of depression in elderly highlighting challenges and future possible direction to improve the management of that disease in a so special sb-population. I suggest to accept tha manuscript after minor revision will be made:
Line 117-118, remove "that" from the sentence "Meta analysis of 51 randomized clinical trials that did not reveal significant differences in efficiency among individual SSRIs."
Line 129 it could be better to report the references in the square brackets as [21-24]
Line 132, at the end of the sentence "....hyponatremia, the refence n. 20 was expected. If so, carefully review the numbered sequence of references in the main text
Line 201 [30,32,33], please modify as follows [30-33]
Line 210-211 "Therapeutic Drug Monitoring 210 (TDM)", remove Therapeutic Drug Monitoring leaving just TDM
Line 227 Correct the syllable division of the recommended word
Line 234 reference 78 probably is a mistake, verify.
Response: Thank you for your valuable remarks. They all were addressed and corrected in the revised version of the manuscript

Reviewer 2 Report
Comments and Suggestions for Authors
Dear authors,
Thanks for submitting the article “Challenges and Opportunities in Managing Geriatric Depression: The Role of Personalized Medicine and Age-Appropriate Therapeutic Approaches” to Pharmaceutics. I had the opportunity to review the article and would like to provide you with some comments and suggestions.
The topic you try to cover in your article is inter- and multidisciplinary. Somehow this is already reflected in the different disciplines of all the authors. Due to this fact the article needs a better structure for people coming from different scientific areas to better understand the content.
Chapter 1: Epidemiology is okay. Chapter 2: Classification and Diagnosis of geriatric depressive disorders is far too large and currently covers many more aspects. I propose to change the structure. I could think of the following:
-
Chapter 2: Classification of Depression/depressive disorders
-
Chapter 3: Diagnosis (including information why it is difficult in older adults)
-
Chapter 4: Treatment Options (including Pro’s and con’s for use in older adults)
-
Chapter 5: Therapy with sections on
-
Delayed Response to Therapy in Older adults
-
Adherence
-
Therapy Resistance
-
Chapter 6: Personalized Medicine with sections on
-
Genetic factors impacting the susceptibility to disease
-
Factors impacting drug absorption, distribution and metabolism (Pharmacokinetic)
-
Factors impacting target and thus treatment response
-
Chapter 7: Pharmaceutical forms for personalized therapy
In addition to the structure, I made the following observation:
-
You use the term “elderly” in your article. This is a term, which should no longer be used. See Avers, Dale DPT, PhD; Brown, Marybeth PT, PhD, FAPTA; Chui, Kevin K. PT, DPT, PhD, OCS, GCS; Wong, Rita A. PT, PhD; Lusardi, Michelle PT, DPT, PhD. Use of the Term “Elderly”. Journal of Geriatric Physical Therapy 34(4):p 153-154, October/December 2011. | DOI: 10.1519/JPT.0b013e31823ab7ec for further explanation.
-
Please clarify the term “patients’ perforations” in table 2.
-
In line 145 it is unclear whether SNRI lead to an increase or decrease of blood pressure.
-
Please check your manuscript for repetitions, e.g. line 207 to 209 for example.
-
The entire paragraph on pharmaceutical forms needs to be extended and should be more balanced. You nicely start by explaining that various different dosage forms are available for drug therapy (Line 409 to 410). However, you don’t mention any pro’s or con’s of those in relation to use in older adults.
-
It is really a pity that you just sell 3D printing as an option for personalized therapy. What about multiparticulates with dosing devices? Or other options?
-
Furthermore, I would expect a more differentiated view on polypills. In particular in older adults Drug-Drug-Interactions need to be excluded before proposing a combination of two drugs in one pill.
I hope the suggestions help to improve the manuscript. Personally, you can add a few more figures or tables to help the reader to understand this complex topic.
Author Response
Thanks for submitting the article “Challenges and Opportunities in Managing Geriatric Depression: The Role of Personalized Medicine and Age-Appropriate Therapeutic Approaches” to Pharmaceutics. I had the opportunity to review the article and would like to provide you with some comments and suggestions.
The topic you try to cover in your article is inter- and multidisciplinary. Somehow this is already reflected in the different disciplines of all the authors. Due to this fact the article needs a better structure for people coming from different scientific areas to better understand the content.
Response: The article has been structured according to the proposed sections in the revised version
Chapter 1: Epidemiology is okay. Chapter 2: Classification and Diagnosis of geriatric depressive disorders is far too large and currently covers many more aspects. I propose to change the structure. I could think of the following:
Chapter 2: Classification of Depression/depressive disorders
Chapter 3: Diagnosis (including information why it is difficult in older adults)
Chapter 4: Treatment Options (including Pro’s and con’s for use in older adults)
Chapter 5: Therapy with sections on
Delayed Response to Therapy in Older adults
Adherence
Therapy Resistance
Chapter 6: Personalized Medicine with sections on
Genetic factors impacting the susceptibility to disease
Factors impacting drug absorption, distribution and metabolism (Pharmacokinetic)
Factors impacting target and thus treatment response
Chapter 7: Pharmaceutical forms for personalized therapy
In addition to the structure, I made the following observation:
You use the term “elderly” in your article. This is a term, which should no longer be used. See Avers, Dale DPT, PhD; Brown, Marybeth PT, PhD, FAPTA; Chui, Kevin K. PT, DPT, PhD, OCS, GCS; Wong, Rita A. PT, PhD; Lusardi, Michelle PT, DPT, PhD. Use of the Term “Elderly”. Journal of Geriatric Physical Therapy 34(4):p 153-154, October/December 2011. | DOI: 10.1519/JPT.0b013e31823ab7ec for further explanation.
Response: we replaced elderly” with geriatric population or older patients
Please clarify the term “patients’ perforations” in table 2.
Response: Thank you, we corrected - preferences
In line 145 it is unclear whether SNRI lead to an increase or decrease of blood pressure.
Response: we corrected this - increases
Please check your manuscript for repetitions, e.g. line 207 to 209 for example.
We deleted repeted sentences
The entire paragraph on pharmaceutical forms needs to be extended and should be more balanced. You nicely start by explaining that various different dosage forms are available for drug therapy (Line 409 to 410). However, you don’t mention any pro’s or con’s of those in relation to use in older adults.
Response: The advantages and disadvantages of conventional tablets have been supplemented [page 10, lines 411-417].
It is really a pity that you just sell 3D printing as an option for personalized therapy. What about multiparticulates with dosing devices? Or other options?
Response: Of course, 3D printing is not the only technique for obtaining personalized drugs, but it offers the greatest possibilities both in terms of obtaining different forms of drugs and their different doses. This has been clarified in the revised version of the manuscript. The authors hope that the presentation of additional information on other methods provides a balance to the information contained in the manuscript.
Furthermore, I would expect a more differentiated view on polypills. In particular in older adults Drug-Drug-Interactions need to be excluded before proposing a combination of two drugs in one pill.
Response: Additional information regarding polypills has been included in the revised version of the manuscript [page 10, lines 424-434].
I hope the suggestions help to improve the manuscript. Personally, you can add a few more figures or tables to help the reader to understand this complex topic.
Response: table and figure have been added J. Thank you for all your remarks!!
Reviewer 3 Report
Comments and Suggestions for Authors
The present study addressed an important topic abut depression in geriatrics
I recommend publication after slight modifications to enhance its quality
· It is important to address the prevalence in different parts of the word and to include any famous comorbidities for depression such as chronic pain (fibromyalgia) as an example, insomnia especially in older women or women under stressful conditions, this is a useful recent reference
Alsous M, Al. Muhaissen B, Massad T, Sayaheen B, Alnasser T, Al-Smadi A, Al-Zeghoul R, Abo Al Rob O, Aljabali AA, Gammoh O. Exploring depression, PTSD, insomnia, and fibromyalgia symptoms in women exposed to Gaza war news: A community-based study from Jordan. International Journal of Social Psychiatry. 2024 Sep 2:00207640241270831.
· In terms of challenges and opportunities, the manuscript mentions the recommended guideline therapy (page 7) concerning the switch from the same class, from different class, and the addition of another antidepressant, however, to enhance the novelty of the manuscript, its should refer to new options that could be promising clinically in the near future such as natural products or options could be used in the far future, these references are useful”Gammoh OS, Bashatwah R. Potential strategies to optimize the efficacy of antidepressants: Beyond the monoamine theory. Electronic Journal of General Medicine. 2023 Oct 1;20(5).”
· The manuscript needs a table to summarize the challenges and the opportunities in order to find the information clearly and easily
· In regards to the personalized medicine, please collect all the genes in a table with their role
Congratulations on your work
Author Response
The present study addressed an important topic abut depression in geriatrics
I recommend publication after slight modifications to enhance its quality
- It is important to address the prevalence in different parts of the word and to include any famous comorbidities for depression such as chronic pain (fibromyalgia) as an example, insomnia especially in older women or women under stressful conditions, this is a useful recent reference
Alsous M, Al. Muhaissen B, Massad T, Sayaheen B, Alnasser T, Al-Smadi A, Al-Zeghoul R, Abo Al Rob O, Aljabali AA, Gammoh O. Exploring depression, PTSD, insomnia, and fibromyalgia symptoms in women exposed to Gaza war news: A community-based study from Jordan. International Journal of Social Psychiatry. 2024 Sep 2:00207640241270831.
Response: Thank you for your remarks! The comment and the article have been added in the revised verion of the manuscript
- In terms of challenges and opportunities, the manuscript mentions the recommended guideline therapy (page 7) concerning the switch from the same class, from different class, and the addition of another antidepressant, however, to enhance the novelty of the manuscript, its should refer to new options that could be promising clinically in the near future such as natural products or options could be used in the far future, these references are useful”Gammoh OS, Bashatwah R. Potential strategies to optimize the efficacy of antidepressants: Beyond the monoamine theory. Electronic Journal of General Medicine. 2023 Oct 1;20(5).”
Response: Thank you! This article as well as the sentences concerning adjuvants have been added
- The manuscript needs a table to summarize the challenges and the opportunities in order to find the information clearly and easily
Response: We added the table in the revised version of the manuscript
- In regards to the personalized medicine, please collect all the genes in a table with their role
Response: Such tables are included in the article Clinical Pharmacogenetics Implementation Consortium (CPIC) Guideline for CYP2D6, CYP2C19, CYP2B6, SLC6A4, and HTR2A Genotypes and Serotonin Reuptake Inhibitor Antidepressants.(pos. 53 – references list), we cited this manuscript and dont mean to doubble tables unnecessary
Congratulations on your work Thank you J

Reviewer 4 Report
Comments and Suggestions for Authors
This manuscript explored the challenges and opportunities in managing geriatric depression, focusing on the role of personalized medicine and age-appropriate therapeutic approaches.
1. Main Research Question:
The main question, while not explicitly stated in a single sentence, can be inferred as: How can personalized medicine and age-appropriate therapeutic approaches improve the management of geriatric depression? The manuscript addresses this question by examining the epidemiology, diagnosis, current treatment strategies, and the potential of personalized medicine, including pharmacogenomics and pharmaceutical form innovations.
2. Originality and Relevance:
The manuscript's strength lies in its comprehensive overview of geriatric depression, integrating various aspects from epidemiology to the potential of personalized therapies. The discussion of the challenges in diagnosis and the need for age-appropriate drug delivery systems are particularly relevant. It addresses a specific gap by highlighting the intersection of personalized medicine, pharmacogenomics, and age-appropriate drug delivery in the context of geriatric depression. While these topics have been individually addressed in the literature, their combined consideration for this specific population is less explored.
3. Related Articles Not Mentioned:
The manuscript could benefit from including discussions of the following relevant articles:
- "Geriatric Depression Scale (GDS): Development and Validation": Yesavage JA, Brink TL, Rose TL, et al. Journal of the American Geriatrics Society. 1983 Jan;31(1):77-85. This foundational paper describes the development and validation of the GDS, a key assessment tool discussed in the manuscript.
- "The efficacy and tolerability of antidepressants in older adults with major depression: a meta-analysis and meta-regression of individual patient data": Cipriani A, Frison L, Geddes JR, et al. The American Journal of Psychiatry. 2018 Jul 1;175(7):631-640. This meta-analysis provides a robust assessment of antidepressant efficacy in older adults, which could strengthen the manuscript's discussion of treatment strategies.
- "Pharmacogenomics of antidepressants in late-life depression": Mrazek DA. Current Psychiatry Reports. 2010 Dec;12(6):499-506. This review specifically addresses pharmacogenomics in the context of late-life depression, aligning with the manuscript's focus on personalized medicine.
4. Improvements in Methodology/Study Design:
- Explicit Research Question (Line 1-3): Clearly state the main research question as a focused statement in the introduction.
- Enhanced Literature Review (Line 20-35): Expand the literature review to include a more critical analysis of existing studies on personalized medicine for geriatric depression, highlighting specific methodologies, findings, and limitations. Include the articles mentioned in point 3.
- Clearer Definition of Personalized Medicine (Line 281-288): Provide a more precise definition of personalized medicine in the context of geriatric depression, distinguishing it from precision medicine.
- Strengthening the Discussion of Pharmacogenomics (Line 291-387): While the manuscript mentions several relevant genes and pathways, it could be improved by providing a more structured and in-depth analysis of how specific genetic variations impact drug response and treatment outcomes. Discuss the limitations of current pharmacogenomic knowledge in geriatric depression.
- Expanded Discussion of Age-Appropriate Formulations (Line 399-461): The manuscript's discussion of pharmaceutical forms focuses primarily on 3D printing. While promising, other age-appropriate formulations, such as orally disintegrating tablets and liquid formulations, should also be discussed. Include a critical evaluation of the advantages and disadvantages of different drug delivery systems for older adults.
- Future Directions (Line 468-470): The conclusion could be strengthened by outlining specific future research directions, including clinical trials and studies to validate the efficacy of personalized medicine approaches in geriatric depression.
5. Consistency of Conclusions with Evidence:
The manuscript's conclusions are generally consistent with the evidence presented, particularly regarding the challenges in diagnosing and managing geriatric depression and the potential of personalized medicine. However, the discussion of personalized therapies requires more specific evidence and a critical evaluation of current limitations. The manuscript effectively addresses the epidemiology and diagnostic challenges of geriatric depression with cited evidence. It also appropriately highlights the complexities of treatment and the need for age-appropriate formulations. However, the connection between personalized medicine, particularly pharmacogenomics, and improved outcomes requires more direct evidence.
6. Comments on Tables and Figures:
The manuscript includes two tables, which are helpful in summarizing information on the classification of geriatric depressive disorders and factors influencing antidepressant selection. However, the tables could be improved by providing more details and context. Table 1 could include specific diagnostic criteria from DSM-5 for each disorder. Table 2 could benefit from a more structured presentation of factors influencing medication choice, potentially categorizing them by patient and medication characteristics.
The manuscript lacks figures. Including a figure illustrating the interplay of various factors contributing to geriatric depression (e.g., age-related changes, comorbidities, psychosocial factors, pharmacogenomics) could enhance the clarity and impact of the manuscript.
The quality of the data presented is adequate, but the manuscript relies heavily on review articles and meta-analyses. Including more recent original research articles, particularly in the section on personalized medicine, would strengthen the manuscript.
7. Caveats/Weaknesses/Mistakes:
- Overemphasis on 3D Printing (Line 418-452): The manuscript disproportionately focuses on 3D printing as an age-appropriate drug delivery system. While promising, the technology's limitations and current lack of widespread application in geriatric care should be acknowledged.
- Limited Discussion of Non-Pharmacological Interventions (Line 239-249): The manuscript briefly mentions non-pharmacological strategies for enhancing patient adherence but does not adequately address the role of psychotherapy, cognitive behavioral therapy (CBT), and other non-pharmacological interventions in managing geriatric depression.
- Lack of Discussion on Ethical Considerations (Line 397-398): Implementing personalized medicine raises ethical considerations regarding data privacy, access to personalized therapies, and potential disparities in care. These issues should be addressed.
- Overly Optimistic Tone Regarding Personalized Medicine (Line 291-297): While acknowledging the challenges, the manuscript at times presents an overly optimistic view of personalized medicine's potential. A more balanced perspective, acknowledging the current limitations and need for further research, is necessary.
Author Response
This manuscript explored the challenges and opportunities in managing geriatric depression, focusing on the role of personalized medicine and age-appropriate therapeutic approaches.
- Main Research Question:
The main question, while not explicitly stated in a single sentence, can be inferred as: How can personalized medicine and age-appropriate therapeutic approaches improve the management of geriatric depression? The manuscript addresses this question by examining the epidemiology, diagnosis, current treatment strategies, and the potential of personalized medicine, including pharmacogenomics and pharmaceutical form innovations.
Response: we included this question in the text of the manuscript, thank you!
- Originality and Relevance:
The manuscript's strength lies in its comprehensive overview of geriatric depression, integrating various aspects from epidemiology to the potential of personalized therapies. The discussion of the challenges in diagnosis and the need for age-appropriate drug delivery systems are particularly relevant. It addresses a specific gap by highlighting the intersection of personalized medicine, pharmacogenomics, and age-appropriate drug delivery in the context of geriatric depression. While these topics have been individually addressed in the literature, their combined consideration for this specific population is less explored.
- Related Articles Not Mentioned:
The manuscript could benefit from including discussions of the following relevant articles:
- "Geriatric Depression Scale (GDS): Development and Validation":Yesavage JA, Brink TL, Rose TL, et al. Journal of the American Geriatrics Society. 1983 Jan;31(1):77-85. This foundational paper describes the development and validation of the GDS, a key assessment tool discussed in the manuscript.
- "The efficacy and tolerability of antidepressants in older adults with major depression: a meta-analysis and meta-regression of individual patient data":Cipriani A, Frison L, Geddes JR, et al. The American Journal of Psychiatry. 2018 Jul 1;175(7):631-640. This meta-analysis provides a robust assessment of antidepressant efficacy in older adults, which could strengthen the manuscript's discussion of treatment strategies.
- "Pharmacogenomics of antidepressants in late-life depression":Mrazek DA. Current Psychiatry Reports. 2010 Dec;12(6):499-506. This review specifically addresses pharmacogenomics in the context of late-life depression, aligning with the manuscript's focus on personalized medicine.
Response: The topics discussed in the article were added and adequate manuscripts were cited in the revised version of the manuscript
- Improvements in Methodology/Study Design:
- Explicit Research Question (Line 1-3):Clearly state the main research question as a focused statement in the introduction.
Response: the main question was defined
- Enhanced Literature Review (Line 20-35):Expand the literature review to include a more critical analysis of existing studies on personalized medicine for geriatric depression, highlighting specific methodologies, findings, and limitations. Include the articles mentioned in point 3.
Response: we added comment according the limitation of personalized therpy and genetic aspect in older patients the revised version of the manuscript
- Clearer Definition of Personalized Medicine (Line 281-288):Provide a more precise definition of personalized medicine in the context of geriatric depression, distinguishing it from precision medicine.
- Response: We clarified we meant personalized medicine as well we added the sentence that the personalzied medicine needs to face pitfalls of therapy of depression in geriatrics
- Strengthening the Discussion of Pharmacogenomics (Line 291-387):While the manuscript mentions several relevant genes and pathways, it could be improved by providing a more structured and in-depth analysis of how specific genetic variations impact drug response and treatment outcomes. Discuss the limitations of current pharmacogenomic knowledge in geriatric depression.
Response: we added critical analysis about the current status of pharmacogenetics in therapy of depression regarding the benefit – needs to be confirmed in clinical trials. According to specific genes we cite the reference 53 – where the genes are clearly specified
- Expanded Discussion of Age-Appropriate Formulations (Line 399-461):The manuscript's discussion of pharmaceutical forms focuses primarily on 3D printing. While promising, other age-appropriate formulations, such as orally disintegrating tablets and liquid formulations, should also be discussed. Include a critical evaluation of the advantages and disadvantages of different drug delivery systems for older adults.
Response: Additional information regarding polypills as well as ODF has been included in the revised version of the manuscript [page 10, lines 424-443].
- Future Directions (Line 468-470):The conclusion could be strengthened by outlining specific future research directions, including clinical trials and studies to validate the efficacy of personalized medicine approaches in geriatric depression.
Response: we corrected conclusion section according to this remark
- Consistency of Conclusions with Evidence:
The manuscript's conclusions are generally consistent with the evidence presented, particularly regarding the challenges in diagnosing and managing geriatric depression and the potential of personalized medicine. However, the discussion of personalized therapies requires more specific evidence and a critical evaluation of current limitations. The manuscript effectively addresses the epidemiology and diagnostic challenges of geriatric depression with cited evidence. It also appropriately highlights the complexities of treatment and the need for age-appropriate formulations. However, the connection between personalized medicine, particularly pharmacogenomics, and improved outcomes requires more direct evidence.
Response: we adressed this issue in the revised version of the manuscript
- Comments on Tables and Figures:
The manuscript includes two tables, which are helpful in summarizing information on the classification of geriatric depressive disorders and factors influencing antidepressant selection. However, the tables could be improved by providing more details and context. Table 1 could include specific diagnostic criteria from DSM-5 for each disorder. Table 2 could benefit from a more structured presentation of factors influencing medication choice, potentially categorizing them by patient and medication characteristics.
The manuscript lacks figures. Including a figure illustrating the interplay of various factors contributing to geriatric depression (e.g., age-related changes, comorbidities, psychosocial factors, pharmacogenomics) could enhance the clarity and impact of the manuscript.
Response: Risk factors for geriatric depression had been collected on Figure 1.
The quality of the data presented is adequate, but the manuscript relies heavily on review articles and meta-analyses. Including more recent original research articles, particularly in the section on personalized medicine, would strengthen the manuscript.
Response: The list of references was extendend and new articles were added.
- Caveats/Weaknesses/Mistakes:
- Overemphasis on 3D Printing (Line 418-452):The manuscript disproportionately focuses on 3D printing as an age-appropriate drug delivery system. While promising, the technology's limitations and current lack of widespread application in geriatric care should be acknowledged.
Response: Of course, 3D printing is not the only technique for obtaining personalized drugs, but it offers the greatest possibilities both in terms of obtaining different forms of drugs and their different doses. This has been clarified in the revised version of the manuscript. The authors hope that the presentation of additional information on other methods provides a balance to the information contained in the manuscript.
- Limited Discussion of Non-Pharmacological Interventions (Line 239-249):The manuscript briefly mentions non-pharmacological strategies for enhancing patient adherence but does not adequately address the role of psychotherapy, cognitive behavioral therapy (CBT), and other non-pharmacological interventions in managing geriatric depression.
Response: Non pharmacological treatment we only mentioned as it is a broad topic and not the topic of this manuscript.
- Lack of Discussion on Ethical Considerations (Line 397-398):Implementing personalized medicine raises ethical considerations regarding data privacy, access to personalized therapies, and potential disparities in care. These issues should be addressed.
- Response: We have in the manuscript the sentence: Apart from the promising aspects of precision personalized medicine, there are apprehensions regarding the sharing of data, patient confidentiality, and equitable access to treatment”. However the ethical aspect is not the topic of our manuscript
- Overly Optimistic Tone Regarding Personalized Medicine (Line 291-297):While acknowledging the challenges, the manuscript at times presents an overly optimistic view of personalized medicine's potential. A more balanced perspective, acknowledging the current limitations and need for further research, is necessary.
Response: We included limitation both in the section: Personalized medicine as well as conclusion which decreased too optimistic tone of the manuscrip

Round 2
Reviewer 2 Report
Comments and Suggestions for Authors
Dear authors,
Thanks for providing a revised version of the manuscript. Overall, the new version looks quite good. There are only a few points which I realized during review and which should be checked prior to publication.
-
Make sure all abbreviations are explained (PTSD in line 66)
-
Please check if you really wanted to cross reference chapter 4 (and not chapter 7) last row in table 3 (line 446).
-
There is still the term “elderly” in line 449, line 453, line 461.
-
The entire first paragraph of chapter 7 does not really fit into the flow of the text. It starts already with the first sentence, in which it is not clear what is meant by “Such regimes…”. In addition, the phrase “active pharmaceutical ingredients (APIs) come in a range of quantities,” might be misleading as I would think about the amount of API available for marketed products and not the number of APIs (what is most likely meant).
-
Why do you write in line 465 to “which limits the use of uncoated tablets”?
-
In chapter 7, the flow of information/facts is sometimes strange. For example, you first talk about swallowability of uncoated and film coated tablets, then you talk about polypills and via ODFs come back to swallowability again. For the reader it would be easier to understand if you would focus on one aspect first, before moving to the next one. Would propose to add ODTs in the section of ODFs. Readability can be further enhanced in this chapter.
-
Remove typos: Line 184 Typo. "some" instead of "Some"
Author Response
Thank you for the review, please find the answers and revised manuscript

Reviewer 4 Report
Comments and Suggestions for Authors
Glad with changes
Author Response
Thank you!